

# Predicting gridded winter PM$_{2.5}$ concentration in east of China

Zhicong Yin[1,2,3], Mingkeng Duan[1], Yuyan Li[1], Tianbao Xu[1], Huijun Wang[1,2,3]

[1]Key Laboratory of Meteorological Disaster, Ministry of Education / Joint International Research Laboratory of Climate and Environment Change (ILCEC) / Collaborative Innovation Center on Forecast and Evaluation of Meteorological Disasters (CIC-FEMD), Nanjing University of Information Science & Technology, Nanjing, 210044, China

[2]Southern Marine Science and Engineering Guangdong Laboratory (Zhuhai), Zhuhai, 519080, China

[3]Nansen-Zhu International Research Centre, Institute of Atmospheric Physics, Chinese Academy of Sciences, Beijing, 100029, China

*Correspondence to*: Zhicong Yin (yinzhc@nuist.edu.cn)

**Abstract.** Exposure to high levels of concentration of fine particle matters with diameter ≤2.5 μm (PM$_{2.5}$) can lead to great threats to human health in east of China. Air pollution control has greatly reduced the PM$_{2.5}$ concentration and entered a crucial stage that required supports like fine seasonal prediction. In this study, we analysed the contributions of emission predictors and climate variability to seasonal prediction of PM$_{2.5}$ concentration. The socioeconomic-PM$_{2.5}$, isolated by atmospheric chemical models, could well describe the gradual increasing trend of PM$_{2.5}$ during the winters of 2001–2012 and the sharp decreasing trend since 2013. The preceding climate predictors have successfully simulated the interannual variability of winter PM$_{2.5}$ concentration. Based on the year-to-year increment approach, a model for seasonal prediction of gridded winter PM$_{2.5}$ concentration (10km×10km) in east of China was trained by integrating of emission and climate predictors. The area-averaged percentage of same sign was 81.8% (relative to the winters of 2001–2019) in the leave-one-out validation. In three densely populated and heavily polluted regions, the correlation coefficients were 0.93 (North China), 0.95 (Yangtze River Delta) and 0.88 (Pearl River Delta) during 2001–2019 and the root-mean-square errors were 6.5, 4.1 and 4.6 μg/m$^3$. More important, the significant decrease in PM$_{2.5}$ concentration, resulted from implementation of strict emission control measures in recent years, was also reproduced. In the recycling independent tests, the prediction model developed in this study also maintained high accuracy and robustness. Furthermore, the accurate gridded PM$_{2.5}$ prediction had the potential to support air pollution control on regional and city scales.



## 1 Introduction

Exposure to fine particle matters with diameter ≤2.5 μm (PM$_{2.5}$) can lead to severe respiratory and cardiovascular diseases (Cohen et al., 2017) and even directly induces DNA damages (Wu et al., 2017). According to the newly recommended air quality guidelines, the level of annual mean PM$_{2.5}$ > 5 μg/m$^3$ has the potential to threat human health (World Health Organization, 2021). In 2020, the average PM$_{2.5}$ concentration in cities of China was 33 μg/m$^3$, although the implementation of strict air quality control measures substantially reduced the emission of primary pollutants (Zhang et al., 2022). The changes in the emission of air pollutants also resulted in the shift of winter PM$_{2.5}$ trend in east of China, that is, the winter PM$_{2.5}$ concentration gradually increased during 2000–2012 but has been decreasing since 2013 (Figure 1a). Evident interannual variation was also be found in the changes of PM$_{2.5}$ concentration in winter, which was largely attributed to climate variability (Yin et al., 2020a, 2020b). Given the severe impact of PM$_{2.5}$ pollution and yearly plan of control action, it is meaningful and urgent to develop prediction models to forecast PM$_{2.5}$ concentration 1~3 months in advance. Furthermore, the predicting results should have high resolution to provide valuable information on the regional and city levels.

To accurately predict climate anomalies is still a real challenge, while predicting air pollution on seasonal scale is much harder than predicting routine meteorological elements (Wang et al., 2021). In general, the methods of climate prediction included numerical climate models and statistical approaches. Despite the great advances in atmospheric chemical models in recent years, most of these models were not designed for real-time operation of seasonal predictions and lacked the coupling of the atmospheric chemical composition and the entire earth system (An et al., 2018). Additionally, statistical prediction of winter PM$_{2.5}$ concentration was limited by the short sequences of observed atmospheric composition, because broad observations only started in 2014 in China. The gray prediction model performed well in dealing with small sample data and thus was used to forecast PM$_{2.5}$ concentration (Wang and Du, 2021; Wu et al., 2019; Xiong et al., 2019). Considering the strong control measures implemented to improve air quality, the buffer operators can be added to the discrete gray prediction model to reduce deviations (Dun et al., 2020). These mathematical models showed certain predictive skills, but lacked of underlying physical mechanisms and long-standing robustness.

Many previous studies employed the long-term observed visibility, air humidity and weather phenomena to reconstruct data of haze (Xu et al., 2016; Zou et al., 2017; He et al., 2019; Yin et al., 2020b). The change in winter haze days consists of long-term trend and interannual-decadal variations. The long-term trend of haze was mainly determined by human activities (i.e., primary pollutants emission and climate change), while its interannual-decadal variations had close relationships with climate variability (Yin et al., 2020b; Geng et al., 2021a). Besides analysis of climate mechanisms, the number of haze days was also used as a proxy-predictand of PM$_{2.5}$ pollution. Taking advantage of the memory effect in slow-varying climate forcings (e.g., sea surface temperature and sea ice), the number of haze days was successfully predicted in North China (Yin and Wang 2016; Yin et al., 2017), Yangtze River Delta (Dong et al., 2021) and Fenwei Plain (Zhao et al., 2021). Chang et al. (2021) used regional stratospheric warming over northeastern Asia in November to predict haze pollution in the Sichuan





Basin in 5–7 weeks. Information from the preceding autumn El Niño was also extracted to predict winter haze days in South
China (Cheng et al., 2019) and aerosol optical depth over northern India (Gao et al., 2019). In most of these studies, the
predictand is area-averaged number of haze days, which was a bit different from $PM_{2.5}$ concentration in use and fine spatial
information was missing.
The Tracking Air Pollution (TAP) database combines information from ground observations, satellite retrievals,
emission inventories and chemical transport model simulations based on data fusion. A full-coverage $PM_{2.5}$ reanalysis
dataset with a spatial resolution of $10km \times 10km$ from 2000 until present has been released (Geng et al., 2021b). It becomes
feasible to develop statistical prediction model of $PM_{2.5}$ concentration based on this long-range dataset. Furthermore, as
reviewed by Yin et al., (2022), the predictability of winter haze decreased during 2014–2020, which was mainly attributed to
the disturbances from super-strict emissions reduction in China. Rapid changes in human activities and changes in climate
anomalies both should be considered and included in $PM_{2.5}$ prediction models. This is the major motivation of the present
study to build a climate-emission hybrid model for the prediction of gridded $PM_{2.5}$ concentration in east of China. The
findings of this study have enormous potentials to support fine designs and implements of air pollution control in advance.
**2 Datasets and method**
**2.1 Data**
The monthly sea ice concentration (SI) and sea surface temperature (SST) dataset, with a spatial resolution of $1° \times 1°$,
were provided by the Met Office Hadley Centre (Rayner et al. 2003, https://www.metoffice.gov.uk/hadobs/hadisst/).
Monthly soil moisture (Soilw), snow depth (SD), geopotential height at 500hPa (Z500) and 850hPa (Z850), sea level
pressure (SLP) and 10m wind were extracted from the fifth generation reanalysis product (ERA5) produced by the European
Center for Medium Range Weather Forecasts (Hersbach et al. 2020,
https://cds.climate.copernicus.eu/#!/search?text=ERA5&type=dataset). Annual emissions of ammonia, nitrogen oxide, BOC,
primary $PM_{2.5}$, and sulfur dioxide in China were derived from the MEIC model (http://www.meicmodel.org/;Li et al., 2017).
Hourly site-observed $PM_{2.5}$ concentration during 2014–2020 were also employed in the present study
(https://www.aqistudy.cn/historydata/). The long-term and high-resolution TAP $PM_{2.5}$ concentration dataset during 2000-
2020 can be downloaded from http://tapdata.org (Geng et al. 2021b). The $PM_{2.5}$ reanalysis data were used as training data as
well as test data in the construction of the prediction model, and the observed $PM_{2.5}$ concentration were also applied to verify
the prediction skill of the model.
**2.2 Isolation of socioeconomic-$PM_{2.5}$**
We employed the simulated annual-mean $PM_{2.5}$ concentrations that exclude the meteorological contributions to
represent the impacts of anthropogenic emissions. Compared with direct use of emission inventory of primary pollutants, the





isolated socioeconomic-PM$_{2.5}$ (SE-PM$_{2.5}$) involved both results of emission changes and follow-up physical and chemical
reactions in the air. To remove the meteorological influences from the TAP PM$_{2.5}$ data, we used chemical transport models
and emission inventories to separate the contributions from emission and meteorology changes. Following the approach
proposed by Xiao et al. (2021), we used a 'fix emission' scenario to quantify the impacts of interannual meteorological
variation on PM$_{2.5}$ concentration in Community Multiscale Air Quality (CMAQ) model. Subsequently, a full simulation with
year-by-year emission and meteorology was completed. Differences between the 'fix emission' simulation and the full
simulation were considered to be PM$_{2.5}$ concentrations driven by anthropogenic emissions. This data has been analyzed to
quantify relative influences of different drivers on PM$_{2.5}$-related deaths in China (Geng et al. 2021b).
**2.3 Year-to-year increment prediction**

The year-to-year increment approach is proposed to improve the skill of climate prediction (Wang et al., 2008), in

which the predicted object is not climate anomalies but is the difference between the current and the previous year (DY).
After adding the predicted DY to the observed predictand in the year before, the final predicted results were obtained. Based
on full use of observations in the previous year, the gradually changing trend and inter-decadal components can be well
reproduced. Anthropogenic-natural-forcing predictand could be represented by Y = YS + YC, where YS and YC denoted the
slowly varying socio-economic and climatic components, respectively. In the DY approach, which was expressed by:
$$DY = Y_t - Y_{t-1} = (YS_t + YC_t) - (YS_{t-1} + YC_{t-1}) = (YS_t - YS_{t-1}) + (YC_t - YC_{t-1})$$
where the subscripts *t* and *t-1* indicated the current and the previous years. Before 2013, the difference between
anthropogenic emissions in two adjacent years was small, Yin and Wang (2016) assumed $(YS_t - YS_{t-1}) \approx 0$ and proposed
that DY was mainly influenced by climate variability. However, due to significant reduction of anthropogenic emissions
after the implementation of China's Air Pollution Prevention and Control Action Plan (Zhang and Geng, 2020), the
assumption of $(YS_t - YS_{t-1}) \approx 0$ was no longer completely valid. Therefore, it is meaningful to consider the information of
rapid emission changes and re-build the prediction model (Yin et al., 2022).

(1) Seasonal prediction model based on SE-PM$_{2.5}$ (SP-SE): this prediction model unilaterally emphasized the impacts of

human activities and was trained by DY of SE-PM$_{2.5}$ in each grid.

(2) Seasonal prediction model based on preceding climate variability (SP-CV): this prediction model was highly

focused on the impacts of climate condition and trained by DY of closely related climate factors.

(3) Seasonal prediction model based on both SE-PM$_{2.5}$ and climate (SP-EC): the contributions of emissions and climate

factors are incorporated into one prediction model, i.e., combining the PM$_{2.5}$ DY from SP-SE and SP-CV.

In the leave-one-out cross validation, root-mean-square error (RMSE), relative bias and correlation coefficient (CC)

were calculated. When discussing the CC after the detrending, the linear trend was removed by stages (i.e., winters of 2001-



2011 and 2012-2019). The percentage of the same sign (PSS; same sign means the mathematical sign of the fitted and
observed $PM_{2.5}$ anomalies was the same) was also computed.
**3 Relative contributions of emission and climate predictors**
**3.1 Roles of emission**

Human activities are the major source of haze pollution in east of China (Zhang and Geng, 2020), which implies that a

large proportion of $PM_{2.5}$ concentration is predictable. Particularly, the large reduction of anthropogenic emissions since
2013 determined the decreasing trend of winter $PM_{2.5}$ concentration (Figure 1a). As aforementioned, the socioeconomic-
$PM_{2.5}$ (i.e., SE-$PM_{2.5}$) isolated by CMAQ could well reflect the impacts of human activities and was a potentially effective
predictor for seasonal prediction of $PM_{2.5}$ concentration. As expected, the one-variable linear regression model based on
anomalies of SE-$PM_{2.5}$ successfully reproduced different slopes of trend during 2001–2007, 2008–2013 and 2014–2019, but
the predicted $PM_{2.5}$ concentration varied too smoothly (Figure S1a). Furthermore, the quantities were underestimated when
observed $PM_{2.5}$ concentration increased and overestimated when $PM_{2.5}$ concentration rapidly decreased. To eliminate the
influence of trend shift, we calculated DY of $PM_{2.5}$ and SE-$PM_{2.5}$. Compared with its anomalies, $PM_{2.5}$ DY did not show
significant trend but displayed regularly oscillating characteristic (Figure 1b), and its predictability was much better (Wang
et al., 2008). The SP-SE model was trained by DY of SE-$PM_{2.5}$ in each grid to predict $PM_{2.5}$ DY. After adding the predicted
$PM_{2.5}$ DY to observed $PM_{2.5}$ in the previous year, the final $PM_{2.5}$ concentration was obtained. The CC between predicted and
observed $PM_{2.5}$ was 0.87 during 2001–2019 in the east of China. The underestimated (2001–2007) and overestimated (2014–
2019) values in Figure S1a were largely corrected and interannual variation also appeared in the results of SP-SE prediction
(Figure S1b). The staged trends from the SP-SE model almost overlapped with the observed trends, indicating the model
performed well in capturing the changes of trend (Figure S2).

North China (NC; 34–42 N, 114–120 E), the Yangtze River Delta (YRD; 27–34 N, 117–122 E) and the Pearl River

Delta (PRD; 21.5–25 N, 112–116 E) are three regions that have been experiencing severe $PM_{2.5}$ pollution (Yin et al., 2015).
Thus, the performance of the SP-SE model in NC, the YRD and the PRD were validated separately (Table 1, Figure 2 a-c).
The RMSEs were 12.2, 6.2 and 6.8 $\mu g/m^3$ in NC, the YRD and the PRD, respectively (Table 1). Larger RMSE in NC did not
indicate the SP-SE model performs worse in NC than in the YRD and the PRD, because the mean value of $PM_{2.5}$
concentration was the highest in NC. The relative bias (absolute bias/mean) in NC was 8.5%, which was smaller than that in
the PRD (12.9%). Consistent with its performance in east of China, the SP-SE model also well reproduced the staged trends
in NC, the YRD and the PRD (Figure 2 a-c). However, when the linear trend was removed, the CC between predicted and
observed $PM_{2.5}$ significantly decreases in all the three $PM_{2.5}$-polluted regions (NC: from 0.78 to –0.13; YRD: from 0.88 to –
0.28; PRD: from 0.74 to 0.16). That is, the prediction model trained by the socioeconomic-$PM_{2.5}$ could well predict the





values and staged linear trends. However, it certainly had no ability to simulate the interannual variability of PM$_{2.5}$
concentration.

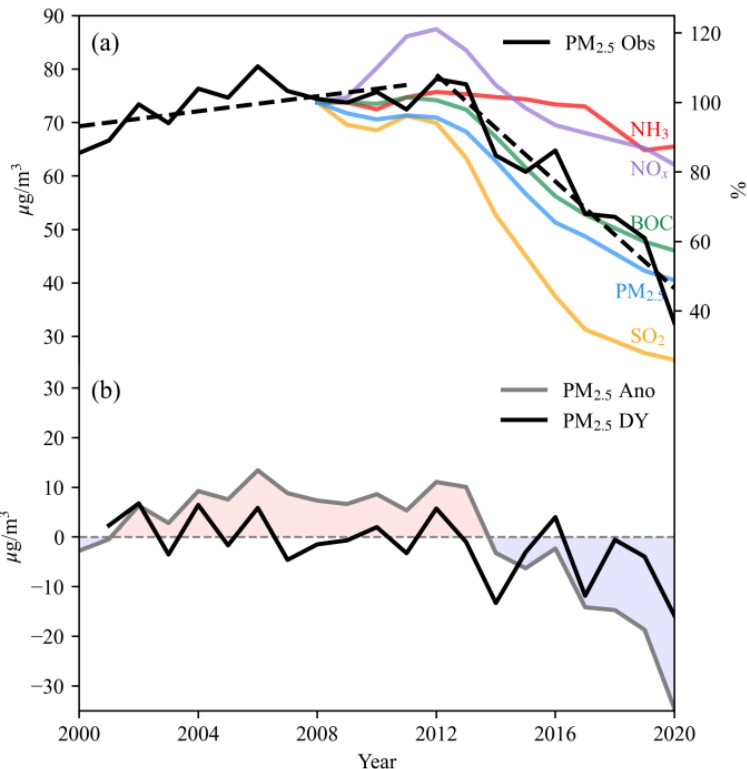


**Figure 1: Variation in (a) winter PM$_{2.5}$ concentration (black; unit: ug/m$^3$), (b) PM$_{2.5}$ anomalies (gray; compared to the mean of**
**2000–2020; unit: ug/m$^3$) and PM$_{2.5}$ DY (black; unit: ug/m$^3$). Color lines in panel (a) indicate relative variations in annual emissions**
**(compared to that in 2008, unit: %) of ammonia (NH$_3$; red), nitrogen oxide (NO$_x$; purple), BOC (green), PM$_{2.5}$ (blue), and sulfur**
**dioxide (SO$_2$; yellow) in east of China. The black dashed line in panel (a) indicates the linear trend of PM$_{2.5}$ concentration.**

**Table 1: The leave-one-out validated root-mean square errors (RMSE), relative biases (absolute bias mean; %) and percentages of**
**same sign (PSS) for three statistical models.**

| | RMSE (µg/m$^3$) | | | Relative Bias (%) | | |
|---|---|---|---|---|---|---|
| | NC | YRD | PRD | NC | YRD | PRD |
| SP-SE | 12.2 | 6.2 | 6.8 | 8.5 | 6.9 | 12.9 |
| SP-CV | 7.6 | 4.7 | 5.2 | 5.2 | 5.9 | 9.7 |
| SP-CE | 6.5 | 4.1 | 4.6 | 4.8 | 5.1 | 8.5 |



**Figure 2: Variations in reanalysis (black) and SP-SE predicted winter PM$_{2.5}$ concentration in (a) NC (orange), (b) the YRD (blue), and (c) the PRD (green) from 2001 to 2019 before (upper) and after (lower) detrending. The predicted PM$_{2.5}$ is dependent on the leave-one-out validation. (d-f) are the same as (a-c), but for SP-CV.**

**3.2 Impacts of climate variability**

Decomposition and prediction of dominant modes of climate conditions were applied in short-term prediction of precipitation (Huang et al., 2022) and surface air temperature (Hsu et al., 2020) in east of China. In this study, we decompose the first four leading modes of PM$_{2.5}$ DY during 2001-2019 (accumulated variance contribution=80.5%) produced by Empirical Orthogonal Function (EOF) analysis, built prediction model for each principal component respectively, recalculate the predicted PM$_{2.5}$ DY by projecting the predicted PCs onto the observed EOF spatial patterns, and finally added the predicted PM$_{2.5}$ DY to the observation in previous year to finish the development of SP-CV (Figure S3, Table S1). The interannual-decadal variation in haze pollution could be well explained by meteorological condition and preceding climate





forcings (Yin et al., 2020b) such as the Arctic sea ice extent (Wang et al., 2015; Yin et al., 2019), Eurasia snow (Zou et al.,
2017) and soil moisture (Yin and Wang 2018), SST in the Pacific (Yin and Wang 2016; He et al., 2019) and Atlantic (Yin
and Zhang 2020a). Prediction signals from these climate anomalies could be observed before winter and owned specific
physical implications.
The first EOF mode of $PM_{2.5}$ DY illustrated heavily haze-polluted status in NC (Figure 3a, e). According to the
correlation analysis, the September SST DY in the Southwest Pacific (CC with PC1=0.76; Figure 4a) and October SST DY
in the Sargasso Sea (CC=0.76; Figure 4b) were selected to be the two predictors for PC1 of $PM_{2.5}$ DY (Table S1). Both of
the predictors had close relationships with dipole pattern of Eurasian anti-cyclonic and Northeast Asian cyclonic circulations
(Figure S4b, c), which was identical to those associated with PC1 (Figure S4a) and could induce cold air from high latitude
to deviate from NC. The second EOF mode indicated a tripole pattern with centers located in the Inner Mongolia, the Fenwei
Plain and South China, respectively (Figure 3b, f). The Fenwei Plain was highly polluted and gained a great attention in
recent years, while the other two centers have relatively better air quality (Zhao et al., 2021). The October snow depth DY in
eastern Siberia (CC with PC2=0.73; Figure 4c), October sea ice DY in the north to Barents Sea (CC=0.75; Figure 4d) and
September-October soil moisture DY in the Indian Peninsula (CC=0.84; Figure 4e) were considered in the prediction model
(Table S1). The predictors possibly induced atmospheric responses in winter (Figure S4 e-g) that were similar to PC2
(Figure S4 d). The anti-cyclonic anomaly over the Fenwei Plain restricted horizontal and vertical dispersion of haze particles
(Zhong et al., 2019).

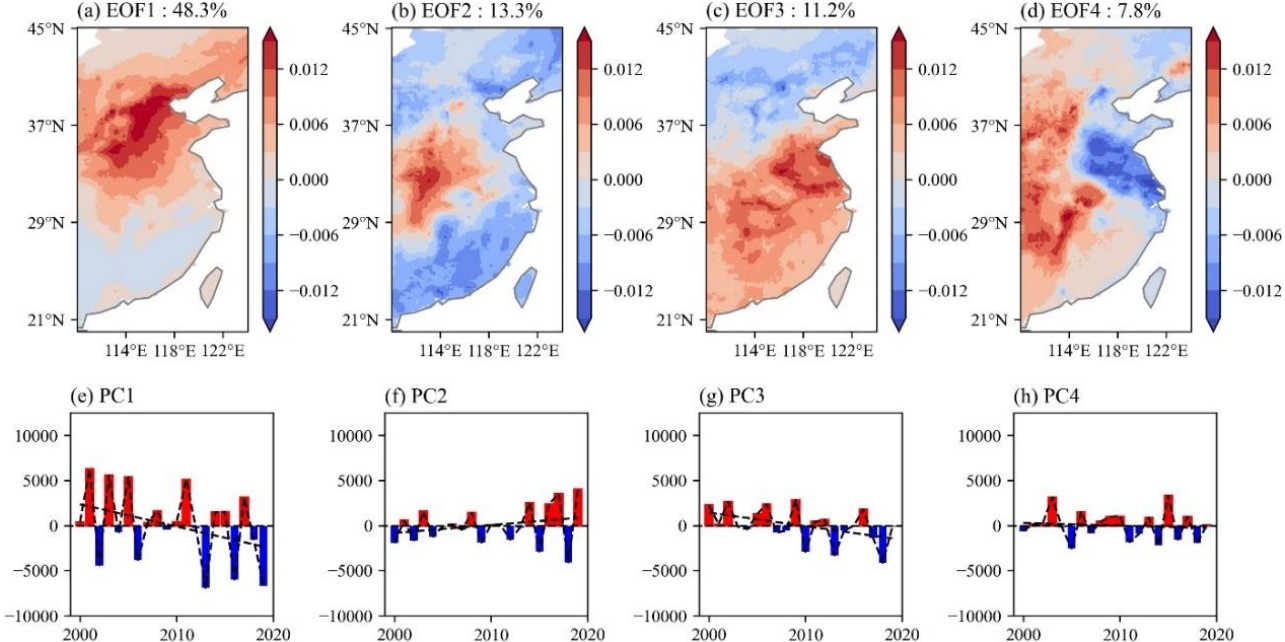

**Figure 3: Spatial patterns (a–d) and corresponding PCs (e–h) of the first four EOF modes for winter $PM_{2.5}$ DY in east of China**
**during 2000–2020. The variance accounted for by each EOF mode is given in the panel.**



**Figure 4: CCs between climate predictors and (a-b) PC1, (c-e) PC2, (g-f) PC3, (h-j) PC4 from 2000 to 2020. The predictors for PC1 are (a) September SST over the South Pacific Ocean and (b) October SST over the Sargasso Sea. The predictors for PC2 are (c) October Sd over eastern Siberia, (d) October SI over the Kara Sea and (e) September-October Soilw over the Indian Peninsula. The predictors for PC3 are (f) October Soilw over the Indo-China Peninsula and (g) June-August SST over the Gulf of Alaska. The predictors for PC4 are (h) October SI over the Chukchi Sea, (i) October soil moisture over the Kamchatka Peninsula and (j) August-September SST over the Arabian Sea and the Bay of Bengal. The slashes indicate CCs exceeding the 95% confidence level. The black boxes indicate the regions over which the predictors are calculated.**





The third EOF mode of PM$_{2.5}$ DY showed a 'north-south' dipole pattern (Figure 3c, g). The variations of PM$_{2.5}$ DY in
Huanghuai and the YRD accounted for a large proportion. The October soil moisture DY in the Indo-China Peninsula (CC
with PC3=0.74; Figure 4f) and June-August SST DY in the Gulf of Alaska (CC=–0.66; Figure 4g) were selected to build
prediction model of PC3 (Table S1). The anomalous atmospheric circulation associated with PC3 and its predictors could
enhance cold air invasion to NC but prevented the cold air from moving further south (Figure S4 h-j). A statistical model
(Table S1) was also developed to predict the 'East-West' dipole shown in the fourth EOF mode (Figure 3d, h) based on
October sea ice DY in the Chukchi Sea (CC=–0.64; Figure 4h), October soil moisture DY in the Kamchatka peninsula
(CC=0.71; Figure 4i) and August-September SST DY in the Arabian Sea (CC=–0.76; Figure 4j). The atmospheric anomalies
in the lower troposphere and near surface, which were associated with the above predictors and PC4, also had similar
impacts on haze pollution (Figure S4 k-n).
As shown in Figure 5, multiple linear regression model demonstrated good performance in simulating the variation in
each PC. The CCs between observed and predicted 1$^{st}$–4$^{th}$ PCs were 0.85, 0.91, 0.79 and 0.93, respectively, all of which
were above the 99% confidence level, indicating that the model successfully reproduced each individual EOF mode.
Meanwhile, the yearly increment approach had the ability to address trend and its changes that were not obviously
mutational (Yin and Wang 2016). The CC between observed and predicted PM$_{2.5}$ concentrations before (after) detrending by
stages was 0.92 (0.64) in NC, 0.94 (0.62) in the YRD and 0.83 (0.59) in the PRD in the leave-one out validation (Figure 2 d-
f). Thus, the SP-CV model well simulated both the trend and the interannual variation of PM$_{2.5}$ concentration in the east of
China. In addition, the RMSEs in NC, the YRD and the PRD were 7.6, 4.7 and 5.3 and the relative biases were 5.2%, 5.2%
and 5.9%, respectively (Table 1), all of which were obviously smaller than those of SP-SE. The PSS, which is an important
indicator of climate prediction, was also evaluated relative to the winters of 2001–2019. The area-averaged PSS from SP-CV
was 80.1% in east of China, which was 8.1% higher than that from SP-SE (Figure 6). Although the SP-CV model performed
better than the SP-SE, especially that it could capture the sharp downward trend after 2013 in NC and YRD, the RMSEs of
the SP-CV simulations for the period 2015-2019 increased up to 11.1, 6.4 and 5.8 μg/m$^3$ in NC, the YRD and the PRD
compared to that of the SP-SE simulations. Obvious positive biases were found in the predictions of PM$_{2.5}$ concentration
after 2014 (Figure 2 d-f) because the SP-CV model was short of information about the super-strict emission regulations
(Figure S2). Based on different levels of haze pollution, various degrees of air pollution control were carried out in NC, the
YRD and the PRD (Zhang and Geng, 2020). In NC, where anthropogenic emissions were most prominently restricted, the
predicted biases were also the largest (Figure 2d). The predicted biases were the smallest in the PRD, while that in the YRD
were in-between. These results were consistent with different intensities of pollution control in the three regions (Figure 2e,
f), which further indicated the importance to fully take into account the impacts of climate variability and anthropogenic
emissions.

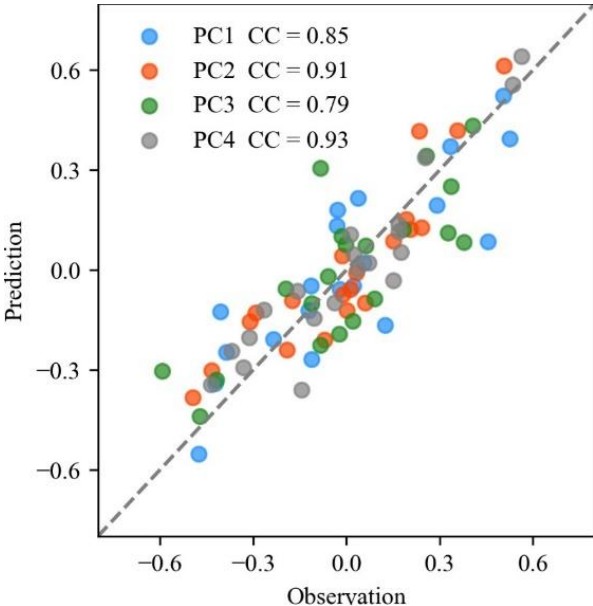

**Figure 5: Scatter plots of observed (x axis) and predicted (y axis) PC1 (blue), PC2 (red), PC3 (green) and PC4 (gray) from 2000 to 2020. The predicted PCs are dependent on the leave-one-cross validation.**

## 4 PM$_{2.5}$ prediction with integrated factors

As aforementioned, the SP-SE model trained by the SE-PM$_{2.5}$ DY considered the impacts of emission changes one-sidedly and could well simulate the values and staged trends. However, it completely failed to reproduce the interannual variation of winter PM$_{2.5}$ concentration in east of China (Figure 2 a-c). Differently, the predictors of climate variability could introduce the interannual variation of winter PM$_{2.5}$ and the yearly increment approach had the ability to bring in the slow trend. The SP-CV model successfully predicted most of the trend and interannual variation in PM$_{2.5}$ concentration (Figure 2 d-f) but underestimated the sharp decreasing trend (Figure S2), which led to positive forecast biases after 2013 (Figure 2d-f).

To fully contain predictive signals of human activities and climate anomalies, the predicted PM$_{2.5}$ DY from SP-SE and SP-CV model for the current year were added up and the sum was added to PM$_{2.5}$ observations in the previous year to develop the final prediction model, i.e., the SP-EC model. As expected, the performance of SP-EC model was better than that of both SP-SE and SP-CV models. Area-averaged PSS was 81.8% in east of China (Figure 6). The CC between observed and SP-EC-predicted PM$_{2.5}$ concentrations before (after) detrending was 0.96 (0.78) in east of China; the RMSE was 2.57 μg/m$^3$, which was 47% (33%) smaller than the RMSE of SP-SE (SP-CV) in the leave-one out validation. That is, the trend simulated by the SP-EC model almost overlapped with the trend of observations (similar to results of SP-SE) and the interannual variation was also reproduced (similar to results of SP-CV). The CCs between observed and SP-EC-predicted PM$_{2.5}$ concentrations before (after) detrending were 0.93 (0.68) in NC, 0.95 (0.44) in the YRD and 0.88 (0.66) in the PRD.



The RMSEs were 6.5 in NC, 4.1 in YRD and 4.6 μg/m$^3$ in PRD, which were 46.7% (14.5%), 33.9% (12.8%) and 32.4%
(11.5%) smaller than that of SP-SE (SP-CV), indicating greater improvements in NC than in the other two regions (Table 1).
According to the relative biases, the SP-EC model also demonstrated a better skill in NC (4.8%) than that in the YRD (5.1%)
and the PRD (8.5%) in the leave-one out validation. As shown in Figure 7, the decreases in PM$_{2.5}$ resulted from the
implementation of strict emission control measures in recent years were also reproduced by the SP-EC model. The evident
and positive biases in the SP-CV results were largely corrected in east of China, NC, the YRD and the PRD (Figure 7).

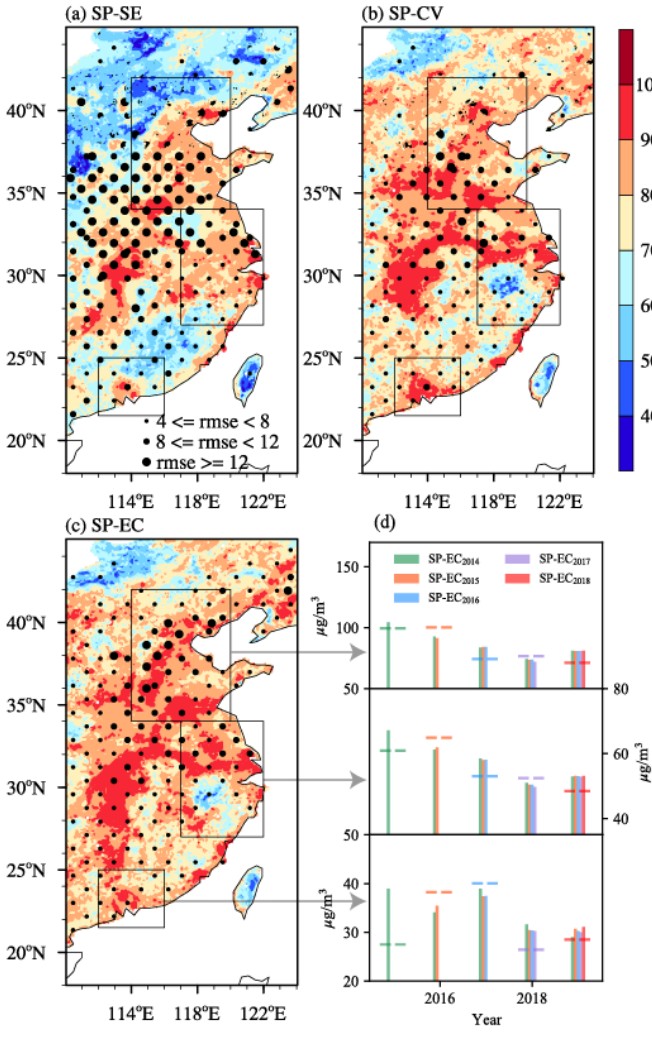


**Figure 6: Distributions of PSS (shadings) and RMSE (dots) from (a) SP-SE, (b) SP-CV, and (c) SP-EC. The boxes represent NC,**
**the YRD and the PRD respectively, and the arrows point to the SP-EC predicted PM$_{2.5}$ in recycling independent tests (bars) and**
**observations (dashed lines) corresponding to the area. The subscript in the legend of panel (d) indicates the model trained from**
**2000 to this year, and the PM$_{2.5}$ from the next year to 2019 are independently predicted.**



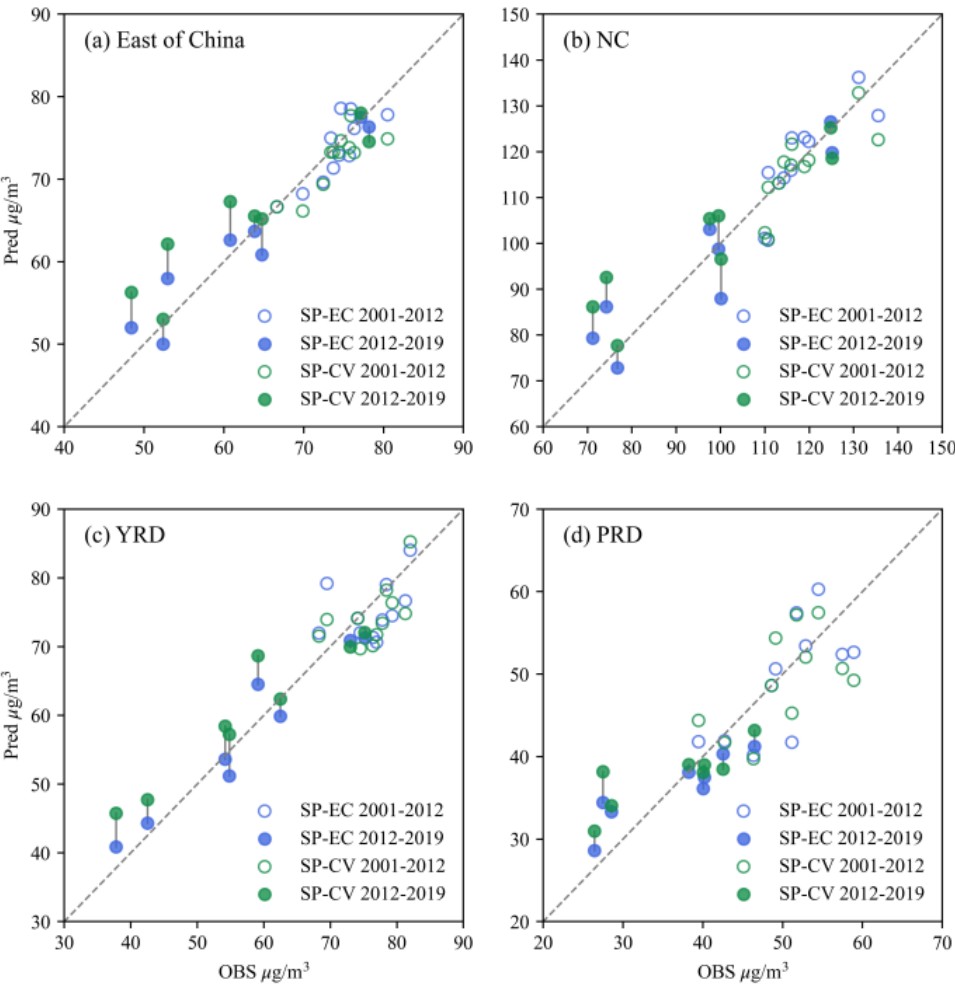

**Figure 7: Scatter plots of the reanalysis (x axis) and predictions of (y axis) PM$_{2.5}$ concentration by SP-CV (green) and SP-EC (blue) in (a) east of China, (b) NC, (c) the YRD and (d) the PRD. The points during 2012–2019 are filled and the short lines between SP-CV and SP-EC pointes indicate the calibrations.**

High spatial resolution was one of the advantages of the seasonal prediction model developed in this study. That is, the SP-EC model could predict winter PM$_{2.5}$ concentration at each 10km×10km grid in east of China. When only considering emission predictors (i.e., SP-SE), RMSEs>12 μg/m$^3$ were found in middle part of the study region and the PSS was lower than 60% in South China and the Inner Mongolia (Figure 6a). When only considering climate predictors (i.e., SP-CV), RMSEs>12 μg/m$^3$ existed in Beijing and its surrounding areas and PSS significantly increased compared to the result of SP-SE (Figure 6b). When integrating both of the emission predictors and climate predictors (i.e., SP-EC), the RMSE in each grid further decreased and the PSS also increased (Figure 6c). In middle part of the study region, the PSS was higher than 80%. In view of gaps between site observations and model simulations, the SP-EC-predicted PM$_{2.5}$ concentrations were



compared with site observations (Figure 8). NC was the most severely polluted area and the SP-EC model could capture the
PM$_{2.5}$ values and interannual differences. Particularly, the SP-EC model reproducee the sudden rebound of PM$_{2.5}$ pollution in
2018 (Figure 8e) that was mainly resulted from climate anomalies (Yin and Zhang 2020a). However, the model failed to
well predict the evident PM$_{2.5}$ drops in east of China (Figure 8f) caused by COVID-19 quarantines (Yin et al., 2021).

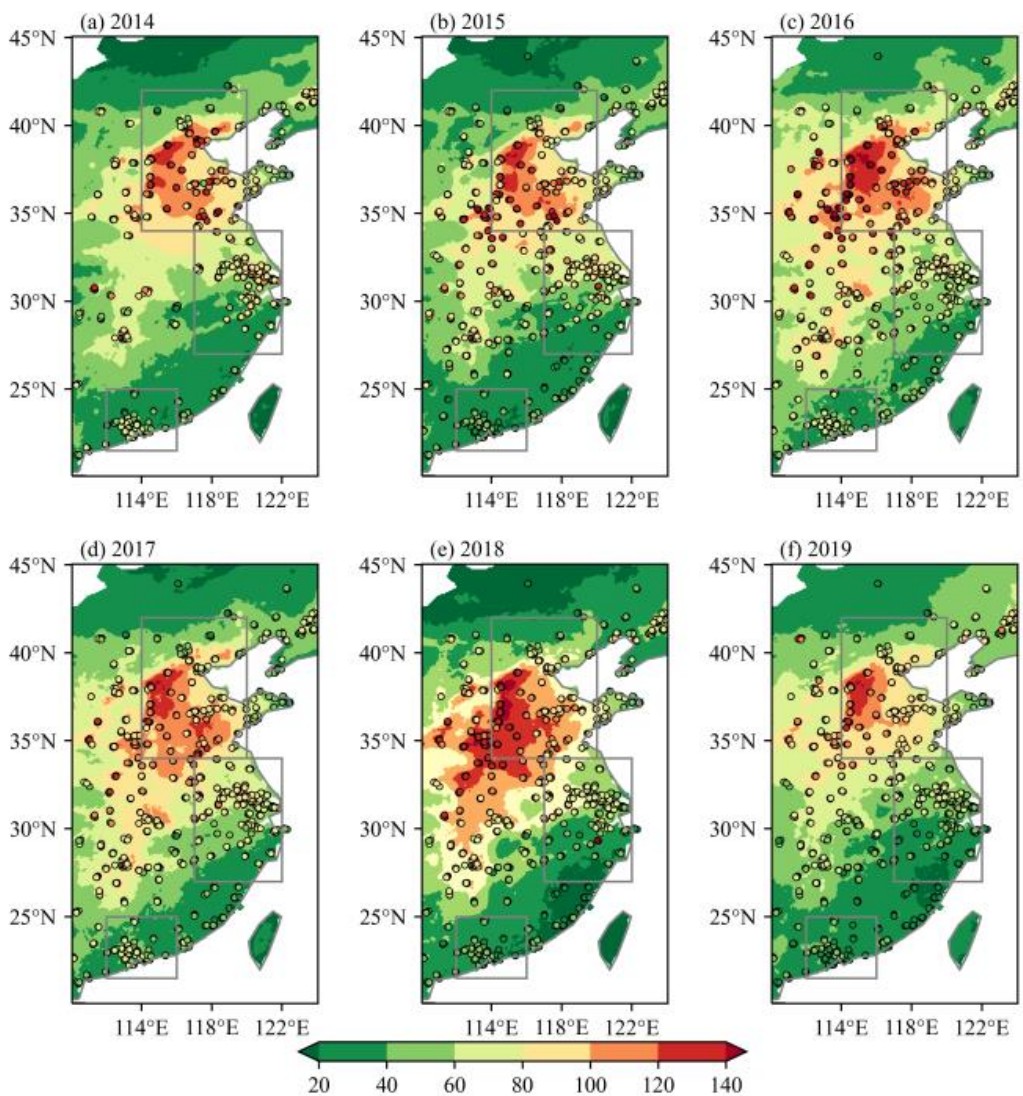


**Figure 8: SP-EC predicted (shading) and site-observed (scatter) PM$_{2.5}$ concentrations in (a) 2014, (b) 2015, (c) 2016, (d) 2017, (e)**
**2018 and (f) 2019. The boxes represent NC, the YRD and the PRD respectively.**
Due to the limitation of short sequence of data, recycling independent tests (RIT) were designed to further verify the
performance of the SP-EC model. In the RIT predictions, the prediction model was trained by samples from 2001 to the
expiration year of training data and the PM$_{2.5}$ anomalies from the next year to 2019 were independently predicted. For





example, the prediction model trained by the data from 2001 to 2014 can produce independent predictions from 2015 to
2019. The expiration year of the training data moved forward from 2015 to 2019, so there were 15 independent predictions.
The PM$_{2.5}$ concentration was independently predicted 5 times for 2019, 4 times for 2018, and so on. The PSS of PM$_{2.5}$
anomalies was 100%, not only relative to winters of 2001–2019 but also 2015–2019, indicating a high accuracy of prediction
in east of China. The predicted values for each year did not vary much (Figure 6d), indicating a high reliability and
robustness of the model. For example, when the SP-EC model was trained by the samples only from 2000 to 2014, the
predicted PM$_{2.5}$ anomalies for 2018 and 2019 were also close to the results of leave-one-out validations and the
measurements.
**5 Conclusions and discussion**

The change of haze pollution consisted of long-term trend, interannual-decadal variations, synoptic disturbances and so
on. Seasonal prediction was focused on predicting long-term trend and interannual-decadal variations 1~3 months in advance
(Wang et al., 2021). Because of the limitation of short observational period, many previous studies employed the number of
haze days as proxy of PM$_{2.5}$ pollution to build statistical prediction model (Yin and Wang 2016; Yin et al., 2017; Dong et al.,
2021; Zhao et al., 2021; Chang et al., 2021). Since 2020, several high-resolution PM$_{2.5}$ reanalysis datasets have been
successively released, which greatly increased the possibility for direct seasonal prediction of PM$_{2.5}$ concentration that is
more familiar to decision makers and the public (Yin et al., 2021).

In this study, two seasonal prediction models were separately trained by emission factor (i.e., SP-SE) or preceding
climate predictors (i.e., SP-CV) to discuss their relative contributions. The SP-SE model could simulate the slow rising trend
of PM$_{2.5}$ concentration before 2012 and the strong downward trend after 2012. However, it was incapable of importing the
interannual component. The SP-CV model benefited from the year-to-year increment approach and could introduce a large
portion of the linear trend except the sharp decrease of winter PM$_{2.5}$ concentration from 2013. Furthermore, the SP-CV
model performed well in predicting the obvious interannual variation of PM$_{2.5}$ concentration. We integrated the emission and
climate factors to establish the final prediction model (i.e., SP-EC), which could well reproduce both the trend and the
interannual variation of PM$_{2.5}$ concentration. The area-averaged PSS was 81.8% in east of China and CC between observed
and predicted PM$_{2.5}$ concentrations before (after) the detrending was 0.96 (0.78). The RMSEs were 6.5 in NC, 4.1 in the
YRD and 4.6 μg/m$^3$ in the PRD, which were 46.7% (14.5%), 33.9% (12.8%) and 32.4% (11.5%) smaller than that the results
of SP-SE (SP-CV). Due to the implementation of the super-strict emission control measures, the air quality has been
substantially improved and this improvement was also perfectly predicted by the SP-EC model. During recycling
independent tests, the PSS of PM$_{2.5}$ anomalies was 100%, demonstrating high accuracy and robustness. The high-resolution
PM$_{2.5}$ prediction could provide scientific supports for air pollution control at the regional and city levels.





This study mainly focused on the development of seasonal prediction model. Although the SP-EC model was proved to
be skilled, the underlying physical mechanisms of climate predictors were not sufficiently explained and needed further in-
deep studies. Furthermore, although the SP-EC model had high spatial resolution, it could only output winter-mean $PM_{2.5}$
concentration. It was meaningful to build monthly models to provide more detailed predictions. In addition, modern weather
and climate forecasts were heavily dependent on numerical prediction models. Thus, it is imperative to design and develop
numerical models that target at routine seasonal prediction of air pollution (Yin et al., 2021). The theories and methods for
seasonal prediction of $PM_{2.5}$ concentration are still exploratory and need further discoveries. Considering the severe impact
of haze pollution, real-time climate prediction is highly demanded for the purpose to determine how to reduce anthropogenic
emissions and how much should be reduced.
**Data availability**
The monthly sea ice concentration and sea surface temperature (SST) dataset were provided by the Met Office Hadley
Centre: https://www.metoffice.gov.uk/hadobs/hadisst/ (Rayner et al. 2003). Monthly soil moisture, snow depth, geopotential
height at 500hPa and 850hPa, sea level pressure and 10m wind were extracted from the fifth generation reanalysis product
(ERA5) produced by the European Center for Medium Range Weather Forecasts:
https://cds.climate.copernicus.eu/#!/search?text=ERA5&type=dataset (Hersbach et al. 2020). Annual emissions of ammonia,
nitrogen oxide, BOC, primary $PM_{2.5}$, and sulfur dioxide in China were derived from the MEIC model:
http://www.meicmodel.org/ (Li et al., 2017). Hourly site-observed $PM_{2.5}$ concentration during 2014–2020 were acquired
from the China National Environmental Monitoring Centre: https://www.aqistudy.cn/historydata/ (CNEMC, 2021). The
long-term and high-resolution TAP $PM_{2.5}$ concentration dataset during 2000-2020 can be downloaded from http://tapdata.org
(Geng et al. 2021b).
**Authors' contribution**
Wang H. J. and Yin Z. C. designed this research. Li Y. Y., Xu T. B. and Duan M. K. performed analyses and trained
prediction models. Yin Z. C. prepared the manuscript with contributions from all co-authors.
**Competing interests**
The authors declare no conflict of interest.
**Acknowledgements**
This research is supported by the National Natural Science Foundation of China (No. 42088101).



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
