# Peer review of "Predicting gridded winter PM2.5 concentration in east of China"

_Atmospheric Chemistry and Physics, 2022_

## Author Comment (AC1)

**Reply to Reviewer 1:**

In this manuscript, the authors integrated the emission and climate factors to establish the prediction model to provide gridded wintertime PM2.5 concentration in east of China. The results showed the model well reproduced both the trend and the interannual variation of PM2.5 concentration. The model also reproduced the significant decrease in $PM_{2.5}$ after the implementation of strict emission control measures since 2013. I acknowledge that the accurate gridded PM2.5 prediction can support air pollution control on regional and city scales. The manuscript is well organized and clearly written, but some details and ambiguous presentation need more clarification. **I recommend a minor revision and my comments are listed below.**

SPECIFIC COMMENTS:

**Can you specify the reason why the study period varies between 2000-2020 and 2001-2019? As DY is the difference between the current and the previous year, the prediction period should be 2001-2020?**

*Reply:*

The varying time range are mainly due to the different valid time range of **different dataset**.

To avoid the confusion, **we have unified it to 2000-2019 and the prediction period is 2001-2019** in the revised manuscript, and more illustrations about time range are added. Minor changes (1 year) do not influence the conclusions of this article.

*Main Revisions **(For brevity, more details are in the revised manuscript)**:*

**Line 21-23:** The area-averaged percentage of same sign was **81.4%** (relative to the winters of 2001–2019) in the leave-one-out validation. In three densely populated and heavily polluted regions, the correlation coefficients were **0.93 (North China), 0.95 (Yangtze River Delta) and 0.87 (Pearl River Delta)** during 2001–2019 and the root-mean-square errors were **6.8, 4.2 and 4.7 μg/m³.**

**Line 78:** The monthly sea ice concentration (SI) and sea surface temperature (SST) dataset from **2000 to 2019**, with…

**Line 85-87:** Hourly site-observed $PM_{2.5}$ concentration during **2014–2019** were also employed in the present study (https://www.aqistudy.cn/historydata/). The long-term and high-resolution TAP $PM_{2.5}$ concentration dataset during **2000-2019** can be downloaded from http://tapdata.org (Geng et al. 2021b).

**Line 104:** After adding the predicted DY to the observed predictand in the year before, the final predicted results **during 2001–2019** were obtained.

**Line 174:** …during 2001-2019 (accumulated variance contribution=**81%**) produced by Empirical Orthogonal Function (EOF) analysis

**Line 183-195:** The first EOF mode of $PM_{2.5}$ DY illustrated heavily haze-polluted status in NC (Figure 3a, e). According to the correlation analysis, the September SST DY in the Southwest Pacific (**CC with PC1=−0.73**; Figure 4a) and October SST DY in the Sargasso Sea (**CC=−0.73**; Figure 4b) were selected to be the two predictors for PC1 of $PM_{2.5}$ DY (Table S1). Both of the predictors had close relationships with dipole pattern of Eurasian cyclonic and Northeast Asian anti-cyclonic circulations (Figure S4b, c), which was identical to those associated with PC1 (Figure S4a) and **could restrain the invasion of cold air from high latitude into NC**. The **second EOF** mode of $PM_{2.5}$ DY showed a 'north-south' dipole pattern (Figure 3b, f). The variations of $PM_{2.5}$ DY in Huanghuai and the YRD accounted for a large proportion. The October soil moisture DY in the Indo-China Peninsula (**CC with PC3=0.73**; Figure 4c) and June-August SST DY in the Gulf of Alaska (**CC=−0.69;** Figure 4d) were selected to build prediction model of PC2 (Table S1). The anomalous atmospheric circulation associated with PC2 and its predictors could enhance cold air invasion to NC (strong northerlies) but prevented the cold air from moving further south (weak 10m winds in Figure S4 d-f).

**Line 207-217:** The **third EOF** mode indicated a tripole pattern with centers located in the east of Inner Mongolia, the Fenwei Plain and South China, respectively (Figure 3c, g). The Fenwei Plain was highly polluted and gained a great attention in recent years, while the other two centers have relatively better air quality (Zhao et al., 2021). The October snow depth DY in eastern Siberia (**CC with PC2=−0.65**; Figure 4e), October sea ice DY in the north to Barents Sea (**CC=−0.60**; Figure 4f) and September-October soil moisture DY in the Indian Peninsula (**CC=−0.79**; Figure 4g) were considered in the prediction model (Table S1). The abnormal northerlies over North China and South China enhanced the horizontal dispersion of haze particles (Zhong et al., 2019), while the weak wind speed and surface wind convergence in central China were conductive to the accumulation of pollutants. A statistical model (Table S1) was also developed to

predict the 'East-West' dipole shown in the fourth EOF mode (Figure 3d, h) based on October sea ice DY in the Chukchi Sea (CC=−0.64; Figure 4h), October soil moisture DY in the Kamchatka peninsula (CC=0.72; Figure 4i) and August-September SST DY in the Arabian Sea (CC=−0.77; Figure 4j). The atmospheric anomalies in the lower troposphere and near surface, which were associated with the above predictors and PC4, also had similar impacts on haze pollution (Figure S4 k-n).

**Line 219:** The CCs between observed and predicted $1^{st}$–$4^{th}$ PCs were **0.82, 0.80, 0.75 and 0.93**, respectively...

**Line 222-223:** The CC between observed and predicted $PM_{2.5}$ concentrations before (after) detrending by stages was **0.91 (0.63) in NC, 0.94 (0.61) in the YRD and 0.83 (0.64) in the PRD** in the leave-one out validation (Figure 2 d-f).

**Line 225-226:** In addition, the RMSEs in NC, the YRD and the PRD were **8.0, 4.8 and 5.2 μg/m³ and the relative biases were 5.3%, 6.2% and 9.9%**, respectively (Table 1),

**Line 227-228:** The area-averaged PSS from SP-CV was **79.9%** in east of China, which was **7.9%** higher than that from SP-SE (Figure 6).

**Line 230-231:** …the RMSEs of the SP-CV simulations for the period 2015-2019 increased up to **11.6, 6.5 and 5.3 μg/m³** in NC, the YRD and the PRD compared to that of the SP-SE simulations.

**Line 252-261:** Area-averaged PSS was **81.4%** in east of China (Figure 6). The CC between observed and SP-EC-predicted $PM_{2.5}$ concentrations before (after) detrending was **0.96 (0.74)** in east of China; the RMSE was **2.7 μg/m³, which was 43.8% (32.5%)** smaller than the RMSE of SP-SE (SP-CV) in the leave-one out validation. That is, the trend simulated by the SP-EC model almost overlapped with the trend of observations (similar to results of SP-SE) and the interannual variation was also reproduced (similar to results of SP-CV). The CCs between observed and SP-EC-predicted $PM_{2.5}$ concentrations before (after) detrending were **0.93 (0.67) in NC, 0.95 (0.42) in the YRD and 0.87 (0.67) in the PRD (Figure 2g-i).** The RMSEs were **6.8 in NC, 4.2 in YRD and 4.7 μg/m³ in PRD, which were 44.3% (15.0%), 32.3% (12.5%) and 30.9% (9.6%) smaller than that of SP-SE (SP-CV),** indicating greater improvements in NC

than in the other two regions (Table 1). According to the relative biases, the SP-EC model also demonstrated a better skill in **NC (5.1%) than that in the YRD (4.9%) and the PRD (8.8%)** in the leave-one out validation.

**Line 314-317:** The area-averaged PSS was **81.4%** in east of China and CC between observed and predicted PM$_{2.5}$ concentrations before (after) the detrending was **0.96 (0.74).** The RMSEs **were 6.8 in NC, 4.2 in the YRD and 4.7 µg/m$^3$ in the PRD, which were 44.3% (15.0%), 32.3% (12.5%) and 30.9% (9.6%) smaller than that the results of SP-SE (SP-CV).**

**Lines 156-159:**

[Figure]

**Figure 1: Variation in (a) winter PM$_{2.5}$ concentration (black; unit: ug/m$^3$), (b) PM$_{2.5}$ anomalies (gray; compared to the mean of 2000–2019; unit: ug/m$^3$) and PM$_{2.5}$ DY (black; unit: ug/m$^3$). Color lines in panel (a) indicate relative variations in annual emissions (compared to that in 2008, unit: %) of ammonia (NH$_3$; red), nitrogen oxide (NO$_x$; purple), BOC (green), PM$_{2.5}$ (blue), and sulfur dioxide (SO$_2$; yellow) in east of China. The black dashed line in panel (a) indicates the linear trend of PM$_{2.5}$ concentration.**

**Lines 161-166:**

**Table 1: The leave-one-out validated root-mean square errors (RMSE), relative biases (absolute bias mean; %) and percentages of same sign (PSS) for three statistical models.**

|  | RMSE ($\mu g/m^3$) | | | Relative Bias (%) | | |
|---|---|---|---|---|---|---|
|  | NC | YRD | PRD | NC | YRD | PRD |
| SP-SE | 12.2 | 6.2 | 6.8 | 8.5 | 6.9 | 12.9 |
| SP-CV | 8.0 | 4.8 | 5.2 | 5.3 | 6.2 | 9.9 |
| SP-EC | 6.8 | 4.2 | 4.7 | 5.1 | 4.9 | 8.8 |

**Lines 167-170:**

[Figure]

**Figure 2: Variations in reanalysis (black) and SP-SE predicted winter PM$_{2.5}$ concentration in (a) NC (orange), (b) the YRD (blue), and (c) the PRD (green) from 2001 to 2019 before (upper) and after (lower) detrending. The predicted PM$_{2.5}$ is dependent on the leave-one-out validation. (d-f) are the same as (a-c), but for SP-CV. (g-i) are the same as (a-c), but for SP-EC.**

**Table 1: "SP-CE" should be "SP-EC".**

*Reply:*

Thank you. We have corrected this error.

*Revisions:*

**Lines 161-166:** Table 1: The leave-one-out validated root-mean square errors (RMSE), relative biases (absolute bias mean; %) and percentages of same sign (PSS) for three statistical models.

| | RMSE ($\mu g/m^3$) | | | Relative Bias (%) | | |
|---|---|---|---|---|---|---|
| | NC | YRD | PRD | NC | YRD | PRD |
| SP-SE | 12.2 | 6.2 | 6.8 | 8.5 | 6.9 | 12.9 |
| SP-CV | 8.0 | 4.8 | 5.2 | 5.3 | 6.2 | 9.9 |
| **SP-EC** | 6.8 | 4.2 | 4.7 | 5.1 | 4.9 | 8.8 |

**Line 188-189: It is hard to find the center located in the Inner Mongolia.**

*Reply:*

Compared of the centers in Fenwei Plain and South China, the center in the east of Inner Mongolia is relatively weak. We have added some "+" and "-" to indicate the centers in Figure 3.

*Revisions:*

**Lines 196-198:**

[Figure]

**Figure 3: Spatial patterns (a–d) and corresponding PCs (e–h) of the first four EOF modes for winter PM₂.₅ DY in east of China during 2000–2019. The variance accounted for by each EOF mode is given in the panel.**

**Line 193: "were similar to PC2" should be "were similar to PC1".**

*Reply:*

We have examined it and this sentence is right.

**Line 210-211: Can you explain more here about how "The anomalous atmospheric circulation associated with PC3 and its predictors could enhance cold air invasion to NC but prevented the cold air from moving further south"?**

*Reply:*

In Figure S4d, both of the anomalous centers of Z500 and SLP were located relatively northward (i.e., to the north of 35°N). Thus, the cold air could move to North China (green arrow) but could not to the south of 35°N (weak wind in the green circles).

Clear explanations are added in the revised version.

[Figure]

**Figure S4d: Correlation coefficients between PC2 and observed DY of atmospheric circulations in winter. The atmospheric variables involved 10m wind, Z500 (contours) SLP (shading). The slashes indicate CCs exceeding the 95% confidence level.**

**Line 224: Please add the units of RMSEs.**

*Reply:*

We have added the units of RMSEs and checked similar contents throughout the manuscript.

*Revisions:*

**Line 225:** In addition, the RMSEs in NC, the YRD and the PRD were **8.0, 4.8 and 5.2 µg/m³ and the relative biases were 5.3%, 6.2% and 9.9%**

**Figure 5: Why is the range of PCs values different from those in Figure 2?**

*Reply:*

The PCs in Figure 5 are **normalized** to plot 4 PCs in one scatter figure. We have revised the caption of Figure 5.

**Figure 8: You should indicate the unit of the shading in the figure or the caption.**

*Reply:*

Thank you. We have added related information in the caption and checked it throughout the manuscript.

*Revisions:*

**Lines 286-287:** Figure 8: SP-EC predicted (shading) and site-observed (scatter) $PM_{2.5}$ concentrations **(units: μg/m³)** in (a) 2014, (b) 2015, (c) 2016, (d) 2017, (e) 2018 and (f) 2019. The boxes represent NC, the YRD and the PRD respectively.

**Line 282-283: "COVID-19 quarantines" occurred in 2020, not in 2019.**

*Reply:*

**Winter is defined as December-January-February** and thus the COVID-19 happened in the winter of 2019 (i.e., January and February in 2020).

We have added the definition of winter and more information.

*Revisions:*

**Lines 38-39:** Evident interannual variation was also be found in the changes of $PM_{2.5}$ concentration in winter (**December-January-February**), which was largely attributed to climate variability (Yin et al., 2020a, 2020b).

**Lines 325-328:** Although the SP-EC model was proved to be skilled……were not sufficiently explained and needed further in-deep studies. As shown in Figure 8f, the SP-EC model failed to well predict the evident $PM_{2.5}$ drops in east of China caused by COVID-19 quarantines **in the winter of 2019 (especially February in 2020)** (Yin et al., 2021).

---

## Author Comment (AC2)

**Reply to Reviewer 2:**

This study by Yin et al. developed a model to predict gridded winter PM$_{2.5}$ concentrations in east of China from a climatological perspective. They integrated both emission and climate variability predictors to train the model, which could capture the trends driven by emission changes and the interannual variations contributed by climate variability. The model has good performance and such method could support air pollution control in the future. **I recommend publication after the following issues are addressed.**

**Line 253-255: A plot similar to Figure 2 but for SP-EC will help the readers to understand the better performance of SP-EC more reasonably.**

*Reply:*

According to the reviewer's suggestion, three panels (g-i) for SP-EC were added in Figure 2.

*Revisions:*

**Lines 167-170:**

[Figure]

**Figure 2: Variations in reanalysis (black) and SP-SE predicted winter PM$_{2.5}$ concentration in (a) NC (orange), (b) the YRD (blue), and (c) the PRD (green) from 2001 to 2019 before (upper) and after (lower) detrending. The predicted PM$_{2.5}$ is dependent on the leave-one-out validation. (d-f) are the same as (a-c), but for SP-CV. (g-i) are the same as (a-c), but for SP-EC.**

**Line 282-283: Figure 8f is for the year 2019, before the COVID-19 quarantine starts. This could hardly be the reason to explain the model biases in this time.**

*Reply:*

**Winter is defined as December-January-February** and thus the COVID-19 happened in the winter of 2019 (i.e., January and February in 2020).

We have added the definition of winter and more information.

*Revisions:*

**Lines 38-39:** Evident interannual variation was also be found in the changes of $PM_{2.5}$ concentration in winter (**December-January-February**), which was largely attributed to climate variability (Yin et al., 2020a, 2020b).

**Lines 325-327:** Although the SP-EC model was proved to be skilled……were not sufficiently explained and needed further in-deep studies. As shown in Figure 8f, the SP-EC model failed to well predict the evident $PM_{2.5}$ drops in east of China caused by COVID-19 quarantines **in the winter of 2019 (especially February in 2020)** (Yin et al., 2021).

I suggest the authors to briefly discuss the uncertainties in this method.

*Reply:*

According to the reviewer's suggestion, the qualitative uncertainties and further studies were briefly discussed.

*Revisions:*

**Lines 324-334:** This study mainly focused on developments of seasonal $PM_{2.5}$ prediction model. **Related theories and methods are still exploratory and need further discoveries**. Although the SP-EC model was proved to be skilled, the underlying physical mechanisms of climate predictors were not sufficiently explained and needed further in-deep studies. As shown in Figure 8f, the SP-EC model failed to well predict the evident $PM_{2.5}$ drops in east of China caused by COVID-19 quarantines in the winter of 2019 (especially February in 2020) (Yin et al., 2021). Therefore, **such sudden fluctuations of $PM_{2.5}$ concentration were not involved in the established prediction model**. Furthermore, **the EOF pattern of $PM_{2.5}$ possibly changed under**

**climate change and must influence the climate component of PM$_{2.5}$**, which should be updated in time. Although the SP-EC model had high spatial resolution, it could only output winter-mean PM$_{2.5}$ concentration. It was meaningful to build **sub-seasonal models** to provide more detailed predictions. Modern weather and climate forecasts were heavily dependent on numerical prediction models. Thus, it is imperative to design and develop numerical models that target at routine seasonal prediction of air pollution (Yin et al., 2021).

**The authors mentioned in the Abstract that the accurate PM2.5 prediction had the potential to support air pollution control on regional and city scales. This worth more discussion in the last section.**

*Reply:*

According to the reviewer's suggestion, we have discussed more about the use of accurate PM$_{2.5}$ prediction.

*Revisions:*

**Lines 319-323:** The high-resolution PM$_{2.5}$ prediction could provide scientific supports for air pollution control at the regional and city levels. For example, real-time PM$_{2.5}$ prediction is highly demanded for determining **how to reduce anthropogenic emissions** and **how much should be reduced**; 10km×10km gridded PM$_{2.5}$ information also had **potentials to support finely and dynamically regional managements and collaborations.**

*SOME TYPOS:*

**Line 166: 'SP-CE' should be 'SP-EC'**

*Reply:*

Thank you. We have corrected this error.

*Revisions:*

**Lines 161-166:** Table 1: The leave-one-out validated root-mean square errors (RMSE), relative biases (absolute bias mean; %) and percentages of same sign (PSS) for three statistical models.

| | RMSE (µg/m$^3$) | | | Relative Bias (%) | | |
|---|---|---|---|---|---|---|
| | NC | YRD | PRD | NC | YRD | PRD |
| SP-SE | 12.2 | 6.2 | 6.8 | 8.5 | 6.9 | 12.9 |
| SP-CV | 8.0 | 4.8 | 5.2 | 5.3 | 6.2 | 9.9 |
| SP-EC | 6.8 | 4.2 | 4.7 | 5.1 | 4.9 | 8.8 |

**Line 271: 'pointes' should be 'points'**

*Reply:*

Thank you. We have corrected this error.

*Revisions:*

**Line 271−273:** Figure 7: Scatter plots of the reanalysis (x axis) and predictions of (y axis) PM2.5 concentration by SP-CV (green) and SP-EC (blue) 269 in (a) east of China, (b) NC, (c) the YRD and (d) the PRD. The **points** during 2012–2019 are filled and the short lines between SP-270 CV and SP-EC points indicate the calibrations.